# LRRK2 dynamics analysis identifies allosteric control of the crosstalk between its catalytic domains

Jui-Hung Weng[1][ᵒ], Phillip C. Aoto[1][ᵒ], Robin Lorenz[2], Jian Wu[1], Sven H. Schmidt[2], Jascha T. Manschwetus[2], Pallavi Kaila-Sharma[1], Steve Silletti[3], Sebastian Mathea[4], Deep Chatterjee[4], Stefan Knapp[4], Friedrich W. Herberg[2], Susan S. Taylor[1]*

1 Department of Pharmacology, University of California, San Diego, California, United States of America, 2 Department of Biochemistry, University of Kassel, Kassel, Germany, 3 Department of Chemistry and Biochemistry, University of California, San Diego, California, United States of America, 4 Institute for Pharmaceutical Chemistry, Goethe University Frankfurt, Frankfurt am Main, Germany

ᵒ These authors contributed equally to this work.
* staylor@ucsd.edu

**Data Availability Statement:** All relevant data are within the paper and its Supporting Information files.

## Abstract

The 2 major molecular switches in biology, kinases and GTPases, are both contained in the Parkinson disease–related leucine-rich repeat kinase 2 (LRRK2). Using hydrogen–deuterium exchange mass spectrometry (HDX-MS) and molecular dynamics (MD) simulations, we generated a comprehensive dynamic allosteric portrait of the C-terminal domains of LRRK2 (LRRK2$_{RCKW}$). We identified 2 helices that shield the kinase domain and regulate LRRK2 conformation and function. One helix in COR-B (COR-B Helix) tethers the COR-B domain to the αC helix of the kinase domain and faces its activation loop, while the C-terminal helix (Ct-Helix) extends from the WD40 domain and interacts with both kinase lobes. The Ct-Helix and the N-terminus of the COR-B Helix create a "cap" that regulates the N-lobe of the kinase domain. Our analyses reveal allosteric sites for pharmacological intervention and confirm the kinase domain as the central hub for conformational control.

## Introduction

Parkinson disease (PD), a major neurodegenerative disorder, is characterized by chronic and progressive loss of dopaminergic neurons. Mutations in the *PARK8* gene, which codes for the leucine-rich repeat kinase 2 (LRRK2), are the most common cause for genetically driven PD [1]. LRRK2 is a large multidomain protein that contains an armadillo repeat motif (ARM), ankyrin repeat (ANK), leucine-rich repeat (LRR), ras-of-complex (ROC) GTPase, C-terminal of ROC (COR), protein kinase, and WD40 domains [2]. While crosstalk between kinases and GTPases, the 2 most important molecular switches in biology, are well-known features in cellular signaling, LRRK2 is one of the few proteins that contains both catalytic domains in the same polypeptide chain [3]. GTP binding to the ROC domain is thought to regulate kinase activity as well as stability and localization [4,5]. Most of the well-known familial mutations are clustered within the ROC, COR, and kinase domains; N1473H and R1441C/G/H in the

**Funding:** This work was supported by Michael J. Fox Foundation Grant 11425 (https://www.michaeljfox.org/) (to S.S.T., and F.W.H.), and Ruth L. Kirschstein National Research Service Award NIH/National Cancer Institute T32 CA009523 (to P.C.A.). JTM was supported by an Otto-Braun Fund Predoctoral Fellowship (B. Braun Melsungen AG). S.M. and S.K. are grateful for support from the Deutsche Forschungsgemeinschaft (DFG) (HE 1818/11) and Structural Genomics Consortium (SGC), a registered charity that receives funds from AbbVie, Bayer Pharma AG, Boehringer Ingelheim, Canada Foundation for Innovation, Eshelman Institute for Innovation, Genome Canada, Innovative Medicines Initiative (875510), Janssen, Merck KGaA Darmstadt Germany, Merck Sharp and Dohme (MSD), Novartis Pharma AG, Ontario Ministry of Economic Development and Innovation. The Synapt G2Si HD/X mass spectrometer was obtained from shared instrumentation NIH Grant S10 OD016234 (to S.S.). The funders had no role in study design, data collection and analysis, decision to publish, or preparation of the manuscript.

**Competing interests:** The authors have declared that no competing interests exist.

**Abbreviations:** A-loop, activation loop; ANK, ankyrin repeat; ARM, armadillo repeat motif; AS, activation segment; COR, C-terminal of ROC; cryo-EM, cryogenic electron microscopy; cryo-ET, cryo electron tomography; Ct-Helix, C-terminal helix; ELK, eukaryotic-like kinase; EPK, eukaryotic protein kinase; GaMD, Gaussian accelerated molecular dynamics; HDX-MS, hydrogen–deuterium exchange mass spectrometry; IBD, inflammatory bowel disease; LRR, leucine-rich repeat; LRRK2, leucine-rich repeat kinase 2; MD, molecular dynamics; NTD, N-terminal domain; PD, Parkinson disease; ROC, ras-of-complex.

GTPase domain and Y1699C in COR-B lie at the interface between the ROC and COR domains, while G2019S and I2020T are in the highly conserved DFGψ motif within the kinase domain [6,7]. This information collectively suggests that there is considerable long-distance communication between the 2 catalytic domains of LRRK2, but can we capture this?

We had previously shown that the kinase domain of LRRK2 is a highly regulated molecular switch. Its conformation regulates more than just kinase activity and plays a crucial role in the intrinsic regulatory processes that mediate subcellular location and activation of LRRK2 [8]. Recent breakthroughs in obtaining structure information, including the in situ cryo-electron tomography (cryo-ET) analysis of LRRK2 polymers associated with microtubules and the high-resolution cryogenic electron microscopy (cryo-EM) structure of the catalytic C-terminal domains (LRRK2$_{RCKW}$), have provided invaluable structural templates that enabled us to achieve a mechanistic understanding of LRRK2 [9,10]. Most recently, the cryo-EM structure of full-length LRRK2 was also solved at high resolution [11].

Here, we combined hydrogen–deuterium exchange mass spectrometry (HDX-MS) and Gaussian accelerated molecular dynamics (GaMD) simulations to gain insight into the dynamic features of LRRK2$_{RCKW}$, a construct that includes both the kinase and GTPase domains. To build a comprehensive allosteric and dynamic portrait of LRRK2$_{RCKW}$, we first mapped our HDX-MS data onto the LRRK2$_{RCKW}$ cryo-EM structure, which gave us a portrait of the solvent accessibility of each peptide. We also assessed the effect of the type I kinase inhibitor MLi-2 and finally used GaMD simulations to monitor the dynamics of LRRK2$_{RCKW}$.

The intrinsic dynamic features of LRRK2$_{RCKW}$ revealed by HDX-MS and GaMD simulations show how the kinase domain is allosterically regulated by its flanking domains. These 2 techniques allow us to explore the molecular features of domain:domain interfaces and loop dynamics. In this way, we identified 2 distinct motifs that control the kinase domain. These 2 motifs, the COR-B helix and the C-terminal helix (Ct-Helix), both impact the overall breathing dynamics of LRRK2$_{RCKW}$. In addition, we showed how the activation segment (AS) of the kinase domain faces the ROC:COR-B interface. This interface is influenced by several PD mutations that cluster in the kinase domain and at the interface between COR-B and the ROC domain. The AS is disordered in the LRRK2$_{RCKW}$ cryo-EM structure, and GaMD simulations allowed us to explore this space. In this inactive conformation, the COR-B Helix is stably anchored onto the αC helix in the N-lobe of the kinase domain, which locks the αC helix into an inactive conformation.

## Results

### Global dynamic portrait of LRRK2$_{RCKW}$ is revealed by HDX-MS and GaMD simulations

To identify the solvent exposed regions of LRRK2$_{RCKW}$, we mapped the HDX-MS data onto the cryo-EM structure of LRRK2$_{RCKW}$ (PDB: 6VNO) [10] (Fig 1A). Our overall HDX-MS coverage of LRRK2$_{RCKW}$, which was >98% (S1 Fig), shows the relative fractional deuterium uptake of each peptide. We previously mapped the HDX-MS profile onto a model of the kinase domain [12] while here we mapped the HDX-MS exchange pattern onto the entire LRRK2$_{RCKW}$ cryo-EM structure. This allows us to capture the interactions between and within the C-terminal domains. As shown in S2A Fig, the C-lobe of the kinase domain around the activation loop (A-loop), a part of the surface region of the ROC domain, and parts of the COR-B domain show the highest deuterium uptake. These regions are either highly flexible or likely to be unfolded, and some are not resolved in the static cryo-EM structure [10]. In contrast, the core of the ROC domain and the core of the COR-B domain show low deuterium uptake suggesting that they are well folded and form rigid domains (S2B Fig). The WD40 domain, except for several loops, also has less deuterium uptake, which indicates that its core is less dynamic in solution.

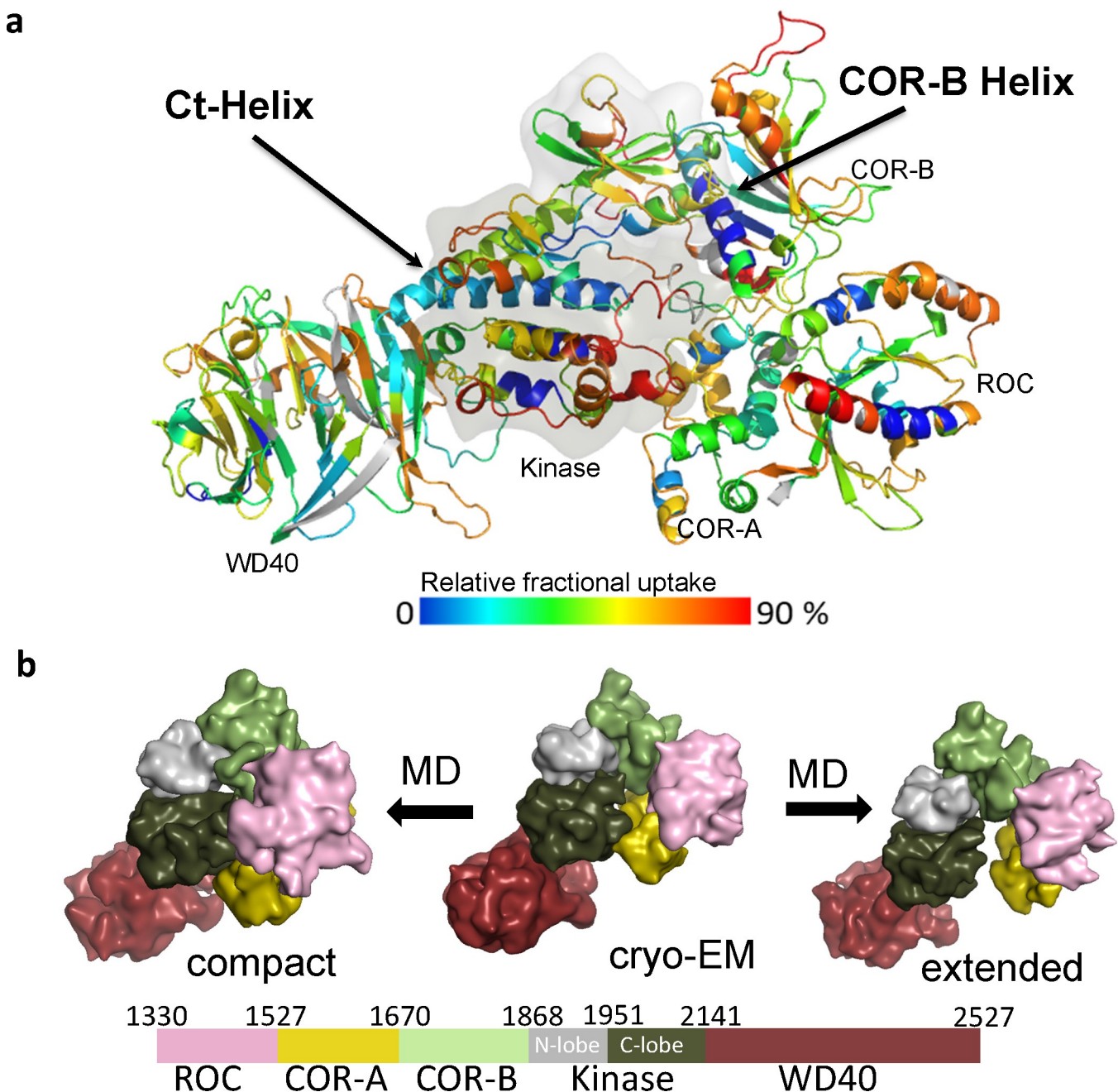

**Fig 1. The overall dynamic of LRRK2_RCKW.** (a) The relative deuterium uptake after 2-min deuterium exposure is color-coded mapped on the LRRK2_RCKW model. Gray color indicates no deuterium uptake information. The surface of the kinase domain is shown in gray. The highly protected COR-B Helix and the Ct-Helix located at the back of the kinase domain are labeled. (b) Snapshot of LRRK2_RCKW in the MD simulation. The surface of each domain is shown in different colors. Left represents the compact architecture of LRRK2_RCKW. The middle is the structure from cryo-EM (PDB: 6VNO) [10]. The right represents one of the extended states of LRRK2_RCKW. cryo-EM, cryogenic electron microscopy; Ct-Helix, C-terminal helix; MD, molecular dynamics; ROC, ras-of-complex.

There are also regions on the surface of each domain that are highly protected from solvent, suggesting that these are domain–domain interfaces. The N-lobe of the kinase domain, for example, is relatively well shielded from solvent in contrast to the highly exposed C-lobe (S2B Fig). The 2 helices (residues 1,567 to 1,578 and 1,600 to 1,612, respectively) of the COR-A

domain at the interface contacting the ROC domain are mostly shielded, indicating that the interaction between COR-A and the ROC domain is stable and persistent. One of the ROC domain helices (residues 1,425 to 1,441) and the adjacent loops of the COR-B domain all have low deuterium uptake, indicating that the ROC:COR-B interface is also shielded from solvent in this inactive conformation. Parts of the kinase domain N-lobe with low deuterium uptake are mostly shielded by the COR-B-Kinase linker and the COR-B domain. Another interface that is well shielded from solvent lies between the kinase and WD40 domain. The beginning of the WD40 domain, the N-terminal end of the αE helix in the kinase domain and the N-terminus of the Ct-Helix, which extends from the WD40 domain, all have very low deuterium uptake, suggesting that the WD40 domain interacts persistently with the C-lobe of the kinase in solution (S2B Fig). This could be one of the reasons why a stable and active isolated kinase domain of LRRK2 has not been expressed yet.

To investigate the dynamic features of LRRK2$_{RCKW}$ and the interactions within its domains, we performed GaMD simulations to recapitulate the behavior of LRRK2$_{RCKW}$ in solution. To capture a more accurate representative model of LRRK2$_{RCKW}$ breathing dynamics, we applied enhanced sampling to broadly sample the conformational changes that take place during the simulations. Both extended and compact conformations of LRRK2$_{RCKW}$ are captured by the simulations (Figs 1B and S3), and the kinase domain is at the center of the breathing dynamics of LRRK2$_{RCKW}$. The COR-B domain persistently interacts with the N-lobe of the kinase during the simulation, while the WD40 domain interacts stably with the C-lobe of the kinase domain. The COR-A and ROC domains move as a single rigid body and fluctuate between far and near states relative to the C-lobe of the kinase domain. When the kinase domain is in a closed conformation, the ROC domain and the COR-A domain are brought closer to the C-lobe of the kinase domain (Fig 1B, left), while in an open conformation (Fig 1B, right), the COR-A domain and ROC domain move further away from the C-lobe of the kinase domain, thereby creating a more extended conformation. The dynamic features that bring the ROC domain and the C-lobe of the kinase domain into close proximity correlate with the intramolecular interactions between the kinase and GTPase domains. As described below, our GaMD simulations also revealed many specific interactions in this space that could potentially be involved in mediating allosteric conformational changes, and these interactions might also be influenced by PD mutations.

## Domain interfaces with the kinase domain

To better understand how the kinase domain is shielded from solvent, we focused on 2 dominant helices that embrace the N- and C-lobes of the kinase domain (S4 Fig). One lies in the COR-B domain (residues 1,771 to 1,791) and is buttressed up against the αC helix in the N-lobe, while the other helix (residues 2,500 to 2,527) lies at the C-terminus and is anchored mainly to the C-lobe of the kinase domain. We refer to these as the COR-B Helix and the Ct-Helix, respectively. In contrast to the well-shielded N-lobe, much of the C-lobe is disordered in the cryo-EM LRRK2$_{RCKW}$ structure (PDB: 6VNO) [10], which represents an inactive conformation. In the kinase domain, we focus on the extended AS, which is typically well ordered in active kinases and poised to interact with substrates and inhibitors [13]. In the LRRK2$_{RCKW}$ structure, the disordered A-loop at the beginning of the AS faces the COR-B Helix, while the region at the end of the AS faces ROC and COR-A (S4 Fig).

## Capturing interactions between the kinase and COR-B domains

The COR-B domain plays a critical role in coordinating the communication that takes place between the kinase domain and the ROC domain. Based on our HDX-MS results (Fig 2 and S1 Table), the interface between the N-lobe of the kinase domain and the COR-B domain is

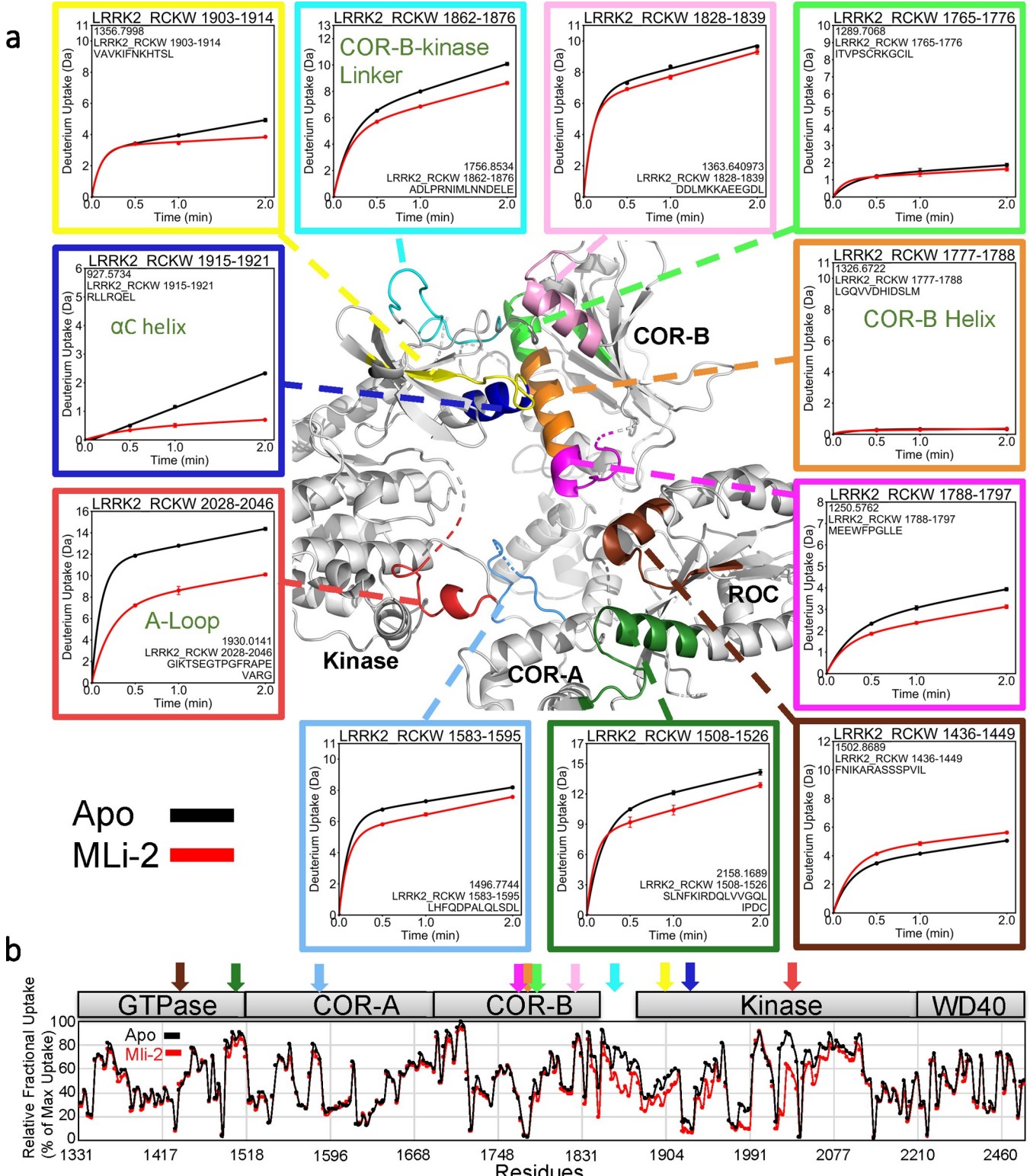

**Fig 2. Deuterium uptake around the COR-B Helix.** (a) The deuterium uptake of selected peptides is plotted and mapped on the LRRK2$_{RCKW}$ structure (PDB: 6VNO). The charts are color coded to the corresponding regions shown. The middle of the COR-B Helix has almost no deuterium uptake, suggesting that it is shielded from the solvent. Other peptides that are located at the surface and the A-loop all demonstrate high deuterium uptake. Binding of MLi-2 reduces the deuterium uptake of COR-B-Kinase linker, the αC helix, and the A-loop. The uptake is also reduced in peptides that are located around the N-terminal or C-terminal ends of the COR-B Helix. Peptide 1,426–1,449 in the ROC domain is the only peptide that its uptake increases when binding to MLi-2. (b) The relative

deuterium exchange for each peptide detected from the N-terminus to the C-terminus of LRRK2$_{RKCW}$ in apo state the kinase (black) and MLi-2 bound (red) conditions at 2 min. The arrows indicate the peptides shown in (a). The data underlying the graphs shown can be found in S1 Data. A-loop, activation loop; ROC, ras-of-complex.

mostly shielded from solvent. This surface is dominated by a long amphipathic helix, the COR-B Helix, and the αC helix of the kinase domain (Fig 2). The COR-B Helix can be divided into 3 segments based on HDX-MS. While the middle region (residues 1,777 to 1,788) shows almost no uptake, the N-terminus shows a slow deuterium exchange. In contrast, the C-terminus of the COR-B Helix is more exposed to solvent but also close to both the disordered A-loop of the kinase domain and to the ROC domain. It has approximately 50% deuterium uptake at 2 min, which is reduced to 35% when the type I LRRK2 kinase inhibitor MLi-2 is bound. The N-terminus of the αC helix in the kinase domain that binds to the COR-B Helix shows an unusual uptake pattern. The uptake increases at a linear rate signifying a slow exchange without reaching a plateau within 2 min, while binding of MLi-2 significantly protects against uptake. Together, HDX-MS shows that the COR-B:kinase domain interface is mostly shielded from solvent and that the COR-B Helix, in particular, is stably anchored to the N-lobe of the kinase while the N-terminus of the COR-B Helix is dynamic and interacts with both the kinase and ROC domains. To dissect these interactions more rigorously, we carefully analyzed the 3 segments of the COR-B Helix that face the αC helix in the kinase domain, the disordered AS and the Ct-Helix (Fig 3A).

The COR-B Helix has charged residues at both its N- and C-termini while the middle part of the COR-B Helix is amphipathic (Fig 3B). Multiple hydrophobic residues face the core of the COR-B domain making this a very stable interface (Fig 3C). On the opposite side, multiple charged residues interact strongly with the αC helix of the kinase domain through electrostatic interactions, locking the αC helix into an inactive conformation (Fig 3C) [14]. Typically, in an active kinase, the basic residue that lies at the beginning of the αC helix (R1915 in LRRK2) interacts with a phosphate on the A-loop, which is missing in this structure [15,16]. Three phosphorylation sites on the A-loop, pT2031, pS2032, and pT2035, have been reported and discussed by others, but how these phosphorylation sites regulate kinase activity is still unclear [17]. Our hypothesis is that the interactions involving the N-terminus of the αC helix will be altered depending on whether the kinase domain is in an active or inactive conformation [14].

The N-terminus of the COR-B Helix interacts with the COR-B-Kinase linker that is more solvent exposed and wraps around the N-lobe of the kinase domain (Fig 3A). This region serves as a "cap" for the N-lobe of the kinase domain, indicating that the conformation and flexibility in particular of the N-lobe appears to be tightly controlled by the linker. The N-terminus of the COR-B Helix, which contains 2 basic residues (R1771 and K1772), also approaches the end of the Ct-Helix that follows the WD40 domain (Fig 4A) and contains the last few residues of LRRK2 that are thought to be crucial for kinase activity. It was previously shown, for example, that deletion of C-terminal residues reduces the kinase activity of LRRK2$_{RCKW}$ [18–20].

In the LRRK2$_{RCKW}$ cryo-EM structure (PDB: 6VNO), the terminal 3 residues (residues 2,525 to 2,527) are disordered [10], while these terminal residues form another turn of the Ct helix in the full-length LRRK2 structure (PDB:7LHW) and in the AlphaFold model [21]. In addition, there is a highly flexible loop (residues 1,715 to 1,730) in the COR-B domain that lies close to the N-lobe of the kinase domain (Fig 5A). This loop is solvent exposed and also disordered in the LRRK2$_{RCKW}$ structure. Our MD simulations show a dominant interaction of the C-terminal residue, E2527, both the side chain and the α-carboxyl group, with R1771 and K1772 at the beginning of the COR-B Helix (Fig 4B).

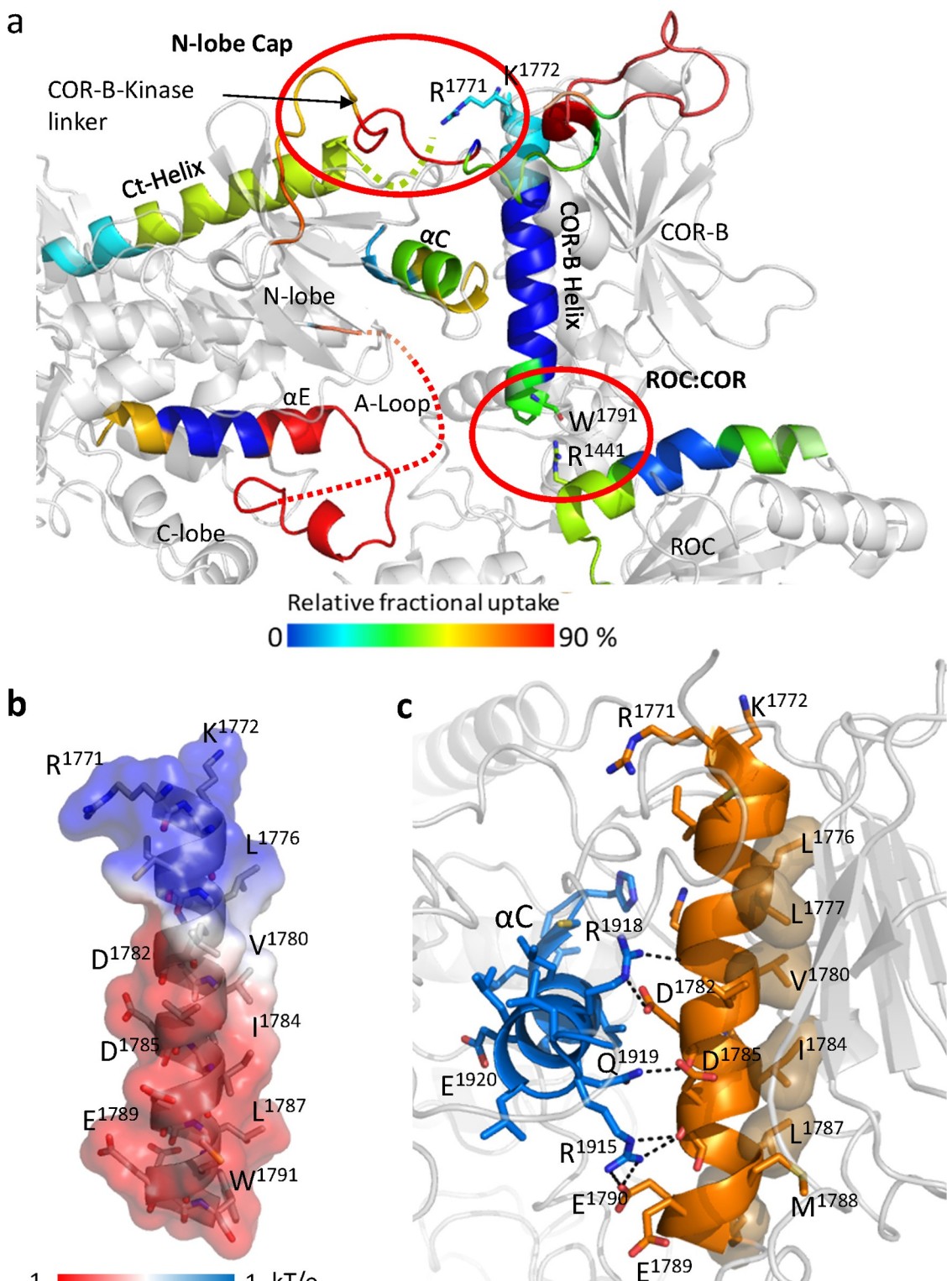

**Fig 3. Characterization of COR-B Helix.** (a) The COR-B Helix, the αC helix and Ct-Helix, and the ROC domain that near the C-terminal end of the COR-B Helix are colored based on the relative fractional uptake (PDB: 6VNO). (b) The surface electrostatic potential of the COR-B Helix. The positively charged N-terminal end of the COR-B Helix is interacting with the Ct-Helix, while the C-terminal end is negatively charged and interacts with the ROC domain. (c) All hydrophobic residues of the COR-B Helix are located on the same side and buried in the COR-B domain, while the charged residues that are forming multiple salt bridges with the αC helix are located on the other side. A-loop, activation loop; Ct-Helix, C-terminal helix; ROC, ras-of-complex.

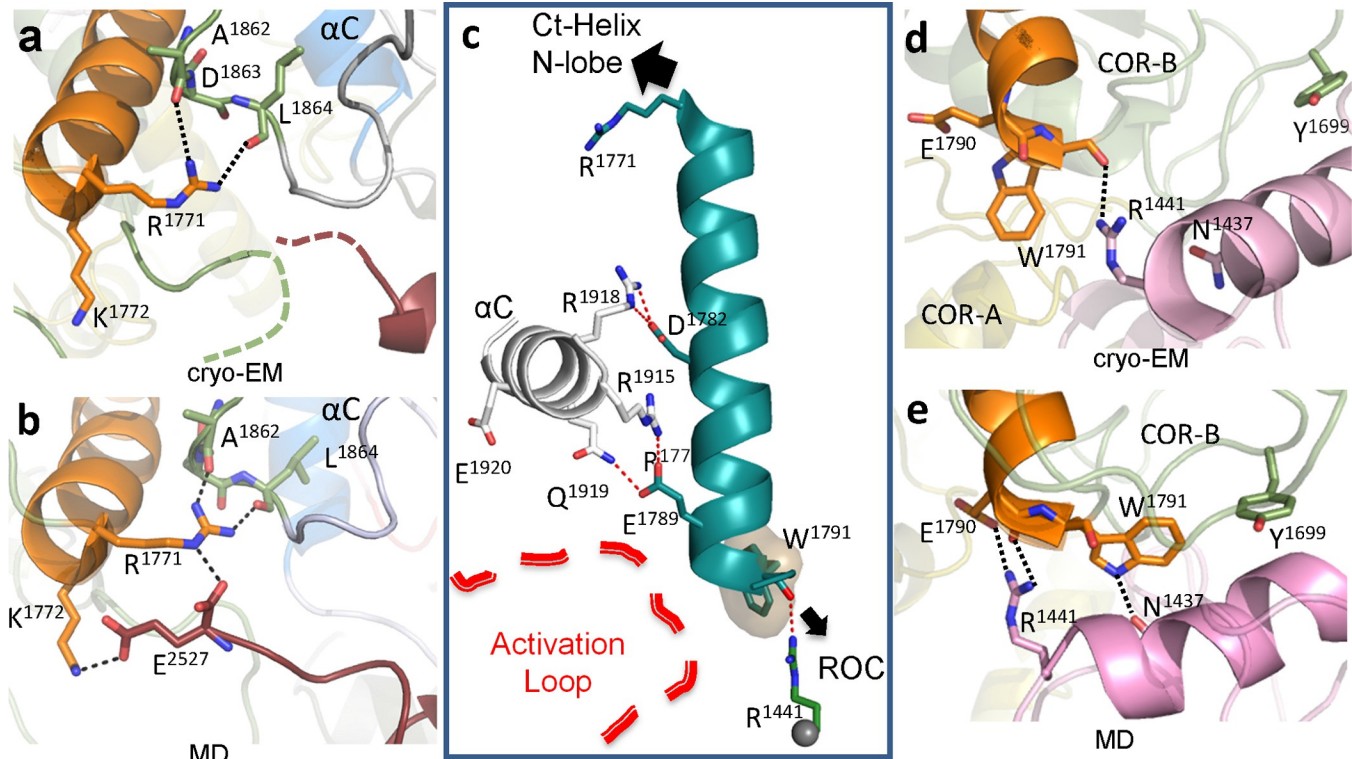

**Fig 4. Capturing interactions at the N-and C-termini of the COR-B Helix.** (a) The N-terminal end of the COR-B Helix is in close proximity to the C-terminal residues of the Ct-Helix, which are undiscernible in the cryo-EM structure (PDB: 6VNO). (b) GaMD simulations capture the interactions between the COR-B Helix with the COR-B-Kinase linker and with the C-terminal end of the Ct-Helix. (c) In LRRK2$_{RCKW}$, the COR-B Helix is stably anchored to the αC helix of the kinase N-lobe. Its N-terminal interacts with the Ct-Helix and the N-lobe of the kinase, while its C-terminal is tethered to the ROC domain and is in close proximity to the A-loop. (d) The C-terminus of the COR-B Helix is anchored to the side chain of R1441 in the ROC domain through W1791. (e) The interaction of E1780 and W1791 with the pathogenic mutation sites R1441 and N1437 on the ROC domain could be captured using GaMD simulations. A-loop, activation loop; cryo-EM, cryogenic electron microscopy; Ct-Helix, C-terminal helix; GaMD, Gaussian accelerated molecular dynamics; MD, molecular dynamics; ROC, ras-of-complex.

## Interactions between the COR-B domain, the ROC domain, and the activation loop in the C-lobe of the kinase domain

The C-terminus of the COR-B Helix is close to both the A-loop of the kinase domain and the ROC domain, and in the inactive conformation that is captured by the cryo-EM structure (PDB: 6VNO), the tip of this helix is anchored to the ROC domain by the side chain of R1441, which binds to the backbone carbonyl of W1791 and helps to "cap" the COR-B Helix (Fig 4C and 4D). The C-terminus of the COR-B Helix also faces the A-loop of the kinase domain, which is likewise disordered in the LRRK2$_{RCKW}$ cryo-EM structure (PDB: 6VNO) (Fig 3A). We know that this is a critical region because exchange of R1441 to either C, G, or H is one of the well-documented PD mutations that leads to activation of LRRK2 [22]. R1441 as well as N1437, another PD mutation at this interface, are also thought to impair the monomer–dimer cycle of LRRK2 and affect GTPase activity [23].

While a single static conformational state is trapped in the cryo-EM structure, GaMD simulations capture additional potential domain:domain interactions that can occur in this region. The simulations suggest, for example, that the side chain of R1441 can also interact with E1790, a residue that is anchored to R1915 in the αC helix of the kinase domain (Figs 3C and 4E and S2 File). This is the residue that would be predicted to interact with the phosphorylation site (P-site) in the A-loop when the kinase is in an active conformation [15,16]. The

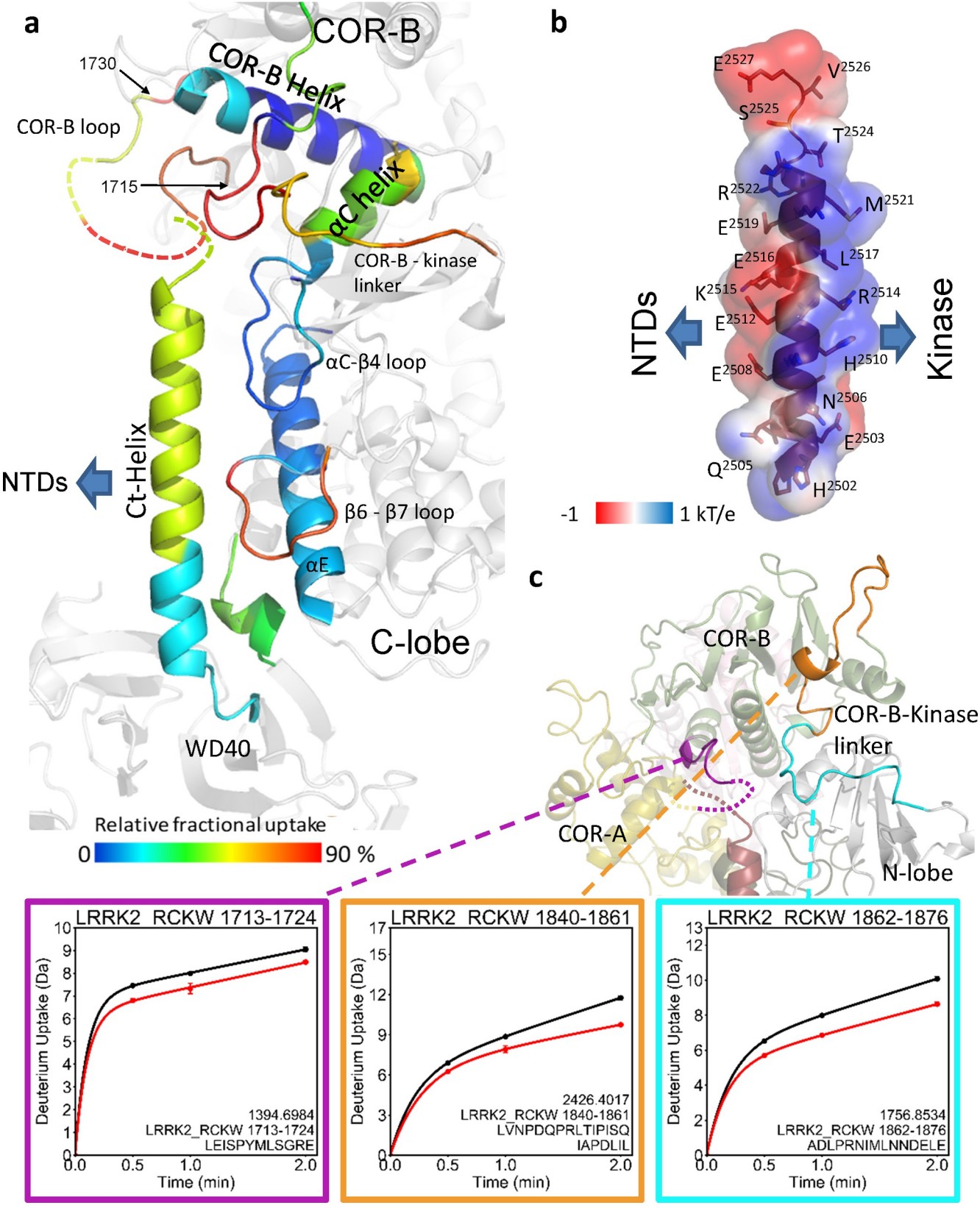

**Fig 5. The dynamic of Ct-Helix.** (a) The Ct-Helix spans across both the N- and the C-lobe of the kinase domain with the C-terminus being located in close proximity to the COR-B Helix and the COR-B loop. The Ct-Helix and the according interaction sites are shown and colored by their relative fractional uptake (PDB: 6VNO). (b) Showing the surface electrostatic potential of the Ct-Helix. The Ct-Helix docks on the kinase domain through the side that is positively charged while the other side is negatively charged and involved in interactions with the NTDs. (c) The deuterium uptake of selected peptides is plotted and mapped on the LRRK2$_{RCKW}$ structure. The COR-B-kinase loop and the loop in COR-B domain both show high deuterium uptake (70%–90%), indicating that they are solvent exposed. And their uptake is reduced in the presence of MLi-2. The data underlying the graphs shown can be found in S1 Data. Ct-Helix, C-terminal helix; NTD, N-terminal domain.

dynamics of the COR-B Helix would likely be significantly influenced by PD mutations of R1441 and N1437, as well as Y1699C, which would all, in principle, uncouple the ROC domain from the COR-B domain and thus enhance potential interactions with the kinase domain that may facilitate activation of LRRK2. Each of these pathogenic mutations would alter the dynamics of the COR-B domain by distinct mechanisms, leaving it free to interact with the A-loop of the kinase domain. Based on our GaMD simulations, multiple residues, such as Q2022, Y2023, and R2026 in the A-loop, can potentially interact with the C-terminus of the COR-B Helix (S5 Fig) once the ROC:COR-B interface is altered in response to PD mutations. These interactions, as discussed later, could be important for stabilizing the A-loop in an extended conformation, which would obviously affect LRRK2 kinase phosphorylation and activation.

## Capturing interactions between the Ct-Helix and the kinase domain

The structures of LRRK2 reveal a unique helix at the C-terminus that extends from the WD40 domain and spans both N- and C-lobes of the kinase domain. This Ct-Helix is present in the inactive LRRK2$_{RCKW}$ cryo-EM structure (PDB:6VNO) and is also docked onto the kinase domain in the full-length cryo-EM structure (PDB:7LHW), which corresponds to an inactive dimer, indicating that it is a very stable helix [9–11]. The combined HDX-MS data captures the dynamic features of the interface of the Ct-Helix with the kinase domain (Fig 5A). The short segment connecting the kinase and the WD40 domain is embedded between the N-terminus of the Ct-Helix and the C-terminus of the αE helix in the kinase domain and all show low deuterium uptake, suggesting a stable interaction between the WD40 domain and the C-lobe of the kinase domain that is shielded from solvent. The Ct-Helix also interacts with the β7-β8 loop in the kinase domain, which is larger in LRRK2 when compared to most kinases (Fig 5A). The αC-β4 loop of the kinase domain, which is an allosteric docking surface for some kinases such as BRAF [24], is almost completely shielded from solvent (S6 Fig and S1 Table). The Ct-Helix interacts with the kinase domain through both hydrophobic and positively charged residues on one side (Fig 5B). The mainly negatively charged residues on the other surface could potentially create a binding interface with the N-terminal domains (NTDs) of LRRK2 when the kinase activity of LRRK2 in the cell is quenched. The recently solved cryo-EM structure of full-length LRRK2 revealed how the ANK and LRR domains interact with the Ct-Helix in the inactive state [11]. The charged surface could also be involved in binding or tethering to substrates or activators when the N-terminal noncatalytic domains are not restrained by the C-terminal domains and in particular by the kinase domain.

## The C-terminal portion of the Ct-Helix directly interacts with the COR-B domain and the N-lobe of the kinase domain

The last 3 residues at the end of the Ct-Helix are at the junction between the COR-B domain and the N-lobe of the kinase domain. This region forms a "cap" for the N-lobe of the kinase domain (Fig 6A). The 2 peptides that cover the COR-B-Kinase linker (residues 1,840 to 1,861 and residues 1,862 to 1,876, respectively) have 70% and 75% uptake of deuterium at 2 min, respectively, and both show a noticeable decrease in deuterium uptake upon binding of MLi-2.

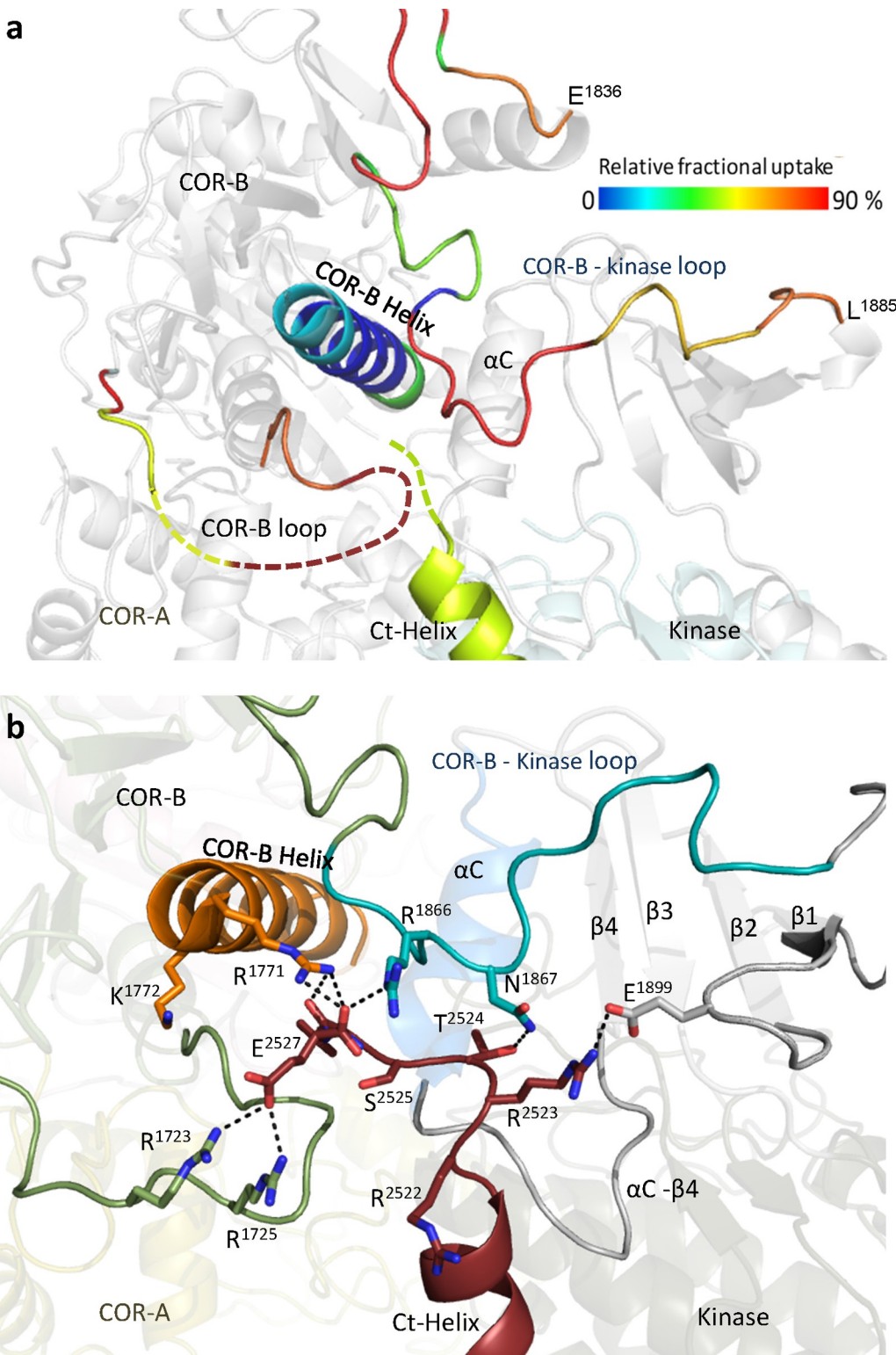

**Fig 6. Capping of the N-lobe of the kinase domain.** (a) In the LRRK2$_{RCKW}$ cryo-EM structure, the linker from COR-B to the kinase domain lies over the N-lobe of the kinase domain. Nearby is a disordered loop from COR-B and the disordered 3 terminal residues (PDB: 6VNO). The loops, COR-B Helix, and the Ct-Helix are colored based on the relative fractional uptake. (b) MD simulations capture potential interactions between the C-terminal residues, the COR-B-kinase loop, and basic residues at the N-terminus of the COR-B Helix. cryo-EM, cryogenic electron microscopy; Ct-Helix, C-terminal helix; MD, molecular dynamics.

The loop in the COR-B domain, which interacts with the C-terminal tail (residues 1,713 to 1,724) also shows reduced uptake (Fig 5C and S1 Table). These changes in deuterium uptake in response to MLi-2 binding indicate correlated changes in dynamics that include both the kinase and COR-B domains and highlights that long-distance communication between the GTPase and kinase domains takes place. Our GaMD simulations capture some of the potential interactions that could occur in this region (Figs 4B and 6B). C-terminal residue E2527 and the free C-terminal carboxyl moiety are of particular interest as the simulations show how the last 5 residues can fluctuate between different structural states and form different interactions with the COR-B domain, the N-lobe of the kinase domain, and the linker between the COR-B domain and the kinase domain. Two arginine residues, R1723 and R1725, in the long loop of the COR-B domain that is disordered in the cryo-EM structures, form H-bonds with E2527 (Fig 6B and S3 File). Another 2 positively charged residues, R1771 and K1772, at the N-terminal end of the COR-B Helix could stabilize the C-terminal tail (residues 2,522 to 2,527) based on the simulations (Fig 4B and S1 File). R1866 on the COR-B-Kinase linker can also bind to E2527. Other residues, such as E1899 in the loop that connects the β2 and β3 strands of the kinase domain, can also interact with R2523 (Fig 6B). In addition, the 2 hydroxyl groups from T2524 and S2525 are also capable of forming H-bonds with either the COR-B or the COR-B-Kinase linker. These interactions that appear in different states during the simulations show how the C-terminal tail can potentially bridge to the COR-B domain and the kinase domain and thereby contribute to the intramolecular network that regulates the kinase and GTPase domains.

T2524 near the C-terminus is a known autophosphorylation site that can be recognized by 14-3-3 binding proteins [25,26]. GaMD simulations with phosphorylated T2524 (pT2524) suggest several new interactions, for example, with R1771 and R1886, when the C-terminal tail is closer to the COR-B-Kinase linker (S7A Fig). When the C-terminal tail is more distant from the N-lobe of the kinase, pT2524 binds to K1772 and R1723 (S7B Fig). The different networks that can be mediated upon phosphorylation can clearly affect the breathing dynamics of LRRK2 and could in turn affect LRRK2 activity. Also, yet to be resolved is whether 14-3-3 binding would stabilize an active or an inactive dimer.

## Capturing the dynamics of the activation loop

As a frame of reference for the LRRK2 AS, we use the AS of the cAMP-dependent protein kinase (PKA) when the A-loop is phosphorylated and the kinase is in a fully closed conformation (S8 Fig). In PKA, the AS begins with the DFGψ motif and ends with the APE motif, 2 of the most highly conserved motifs in the protein kinase superfamily [27]. In between these 2 motifs are the A-loop and the P+1 loop. The APE-αF linker that follows the AS typically plays an important role in docking of substrates and other proteins and should be considered an extended part of the AS [13]. Our HDX-MS results show that the A-loop and part of the P+1 loop in the LRRK2 kinase domain are highly dynamic and likely unfolded (Fig 2 and S1 Movie), which is consistent with the fact that the A-loop and most of the P+1 loop are not resolved in the LRRK2$_{RCKW}$ cryo-EM structure (PDB: 6VNO) (Fig 7A). The rest of the P-loop including the region extending from the APE motif through to the αF helix is, however, folded, and overlays well with the corresponding region of PKA. We consider first the disorder region in the LRRK2$_{RCKW}$ structure. Several key residues in this disorder region face out toward the solvent, with the corresponding residues in PKA (R194) serving as a docking site for the regulatory subunits (R) (S9 Fig). The rest of the P+1 loop through to the αF helix is ordered and in a conformation that resembles PKA, and this conformation is conserved in the full-length LRRK2 structure and in the AlphaFold model. Several conserved residues/motif are embedded

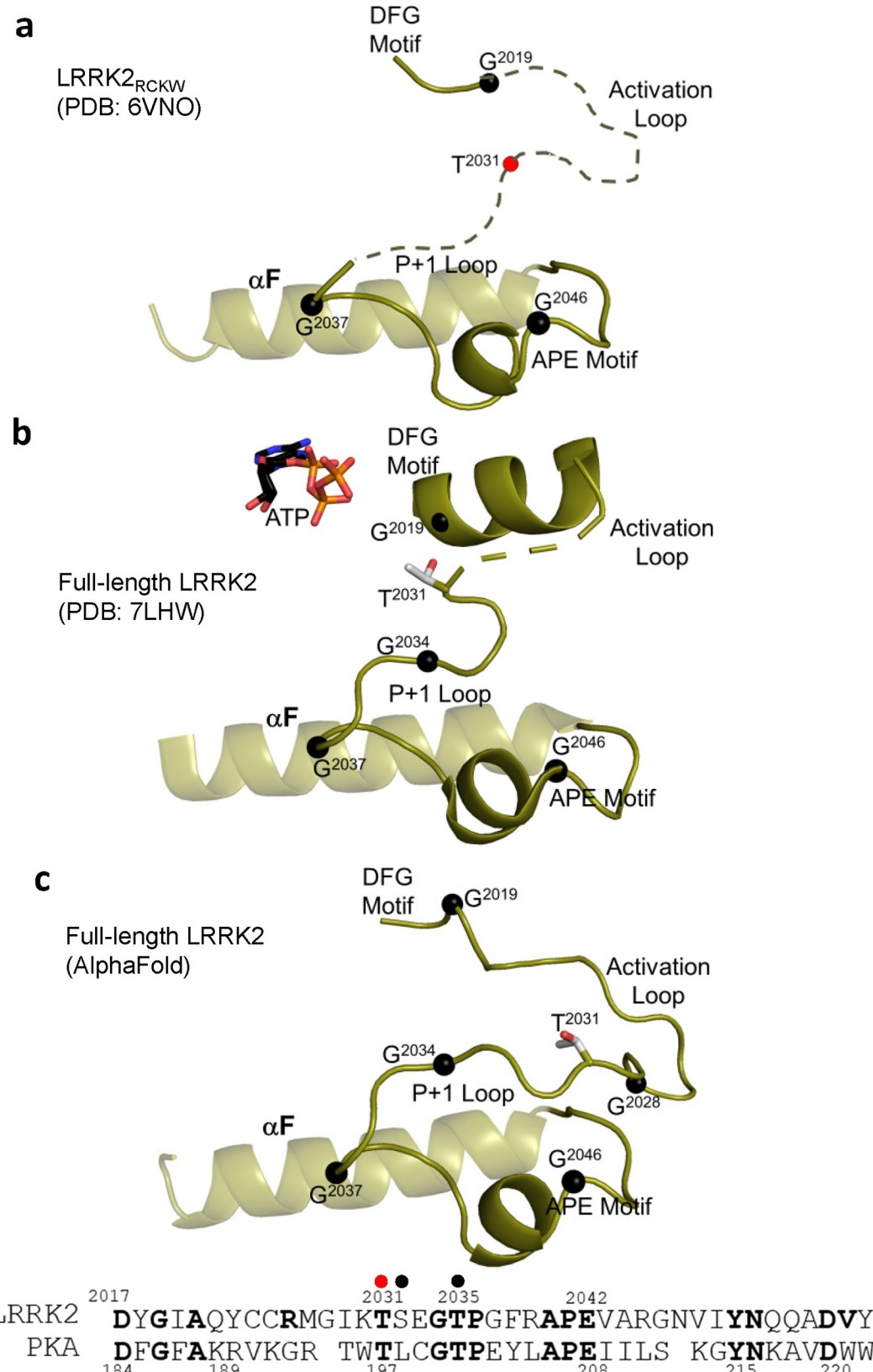

**Fig 7. Comparison of AS of LRRK2.** The motifs that are embedded in the AS of LRRK2$_{RCKW}$ (a), full-length LRRK2 (b), and the model from AlphaFold database (c) are summarized. The AS begins with the DFG motif and ends with the APE motif, 2 of the most highly conserved motifs in the protein kinase superfamily. In between these 2 motifs are the A-loop and the P+1 loop. While the AS in the inactive LRRK2$_{RCKW}$ structure is mostly disordered, the AS in the inactive full-length LRRK2 is mostly ordered, only residue 2,028–2,030 are missing. The A-loop phosphate in LRRK2, T2031, is indicated as a red sphere in panel a and in the sequence alignment, which is the corresponding residue of T197 in PKA. Additional P-sites in LRRK2 are shown as black spheres in the sequence alignment. Black spheres in panels a–c correspond to highly conserved glycines that may serve as hinge points. A-loop, activation loop; AS, activation segment; LRRK2, leucine-rich repeat kinase 2.

in this segment. In PKA, one of these highly conserved residues, Y215 (Y2050 in LRRK2), bridges to the backbone residues of the P-site (pT197) in the AL when the active kinase is phosphorylated on its A-loop [28,29] (S10 Fig). In the cryo-EM structure of LRRK2 (PDB: 6VNO), the AL is not phosphorylated and this Tyr is not in an active-like conformation.

Another interaction in this region, the APE motif, is a critical feature that distinguishes the eukaryotic protein kinases (EPKs) from the eukaryotic-like kinases (ELKs), which are their evolutionary precursors [28]. In the ELKs, the A-loops are short and not dynamic, and the helical domain (αG, αH, αI) is missing. The APE motif in the AS is anchored to this helical domain by a key and highly conserved electrostatic interaction between E2042 in the APE motif of LRRK2 and R2122 in the αH-αI loop. In the LRRK2$_{RCKW}$ cryo-EM structure, these 2 residues are close but not within hydrogen bonding distance (PDB: 6VNO) (S11A Fig); however, the interaction between E2042 and R2122 is frequently captured in the GaMD simulations (S11B Fig) and is actually present in the full-length LRRK2 structure (PDB: 7LHW) (S11C Fig) and in the AlphaFold model (S11D Fig).

Surprisingly, when looking at the inactive cryo-EM structure of full-length LRRK2 (PDB: 7LHW), which also contains ATP, the AS is mostly ordered except for 3 residues (Fig 7B). In this structure, the complete P+1 loop is ordered in a way that agrees with the AlphadFold model (Fig 7C) and overlays well with PKA (PDB: 1ATP). However, the DFGψ motif region is ordered in a helix that is buttressed up against the N-lobe in contrast to an active conformation where the DFGψ motif would be fused to a beta strand that binds to the C-lobe (S10 Fig). Based on PKA, there are at least 2 docking motifs embedded within the AS. One docking site, discussed above, is created by the outward facing surface of the A-loop; the other is created by the outward docking surface of the APE-αF motif. In PKA, these 2 sites are integrated to create a highly dynamic allosteric site that is destroyed by the binding of cAMP (S9 Fig). In LRRK2, based on our HDX-MS/GaMD data, we predict that the A-loop will reach over to the COR-B Helix while the APE-αF linker will dock onto COR-A. Both states are captured in our GaMD simulations (S12 Fig).

## Discussion

The mechanisms that control the intrinsic regulation of LRRK2 include kinase activity as well as targeting to different subcellular sites and the transition between monomeric and dimeric states. In an attempt to capture some of the interdomain interactions, we analyzed the cryo-EM LRRK2$_{RCKW}$ structure [10]. This structure, which represents a static snapshot, was used as our starting point. With HDX-MS and GaMD simulations, we were able to explore more deeply in domain:domain interfaces and loop dynamics, which allowed us to create a dynamic portrait of LRRK2$_{RCKW}$. Based on the solvent-shielded and solvent-exposed regions, we defined 3 rigid bodies and were able to confirm this domain organization using GaMD simulations (S13 Fig). The kinase domain is solidly anchored to the WD40 domain as well as the Ct-Helix that extends from the WD40 domain. The catalytically inert COR domain is comprised

of 2 subdomains, referred to as the COR-A and COR-B domains, joined by a flexible linker. The COR-A domain is firmly anchored to the ROC domain so that these 2 subdomains also move as a rigid body while the COR-B domain functions as a separate rigid body that interacts with both the kinase domain and the ROC domain as well as with the C-terminus (residues 2,525 to 2,527). As predicted by Watanabe and colleagues, this highly dynamic COR-B domain is the major mediator of the communication that takes place between the kinase domain and the ROC domain in the active dimer [9].

GaMD show that domain motions are also embedded within the kinase domain. The kinase domain, for example, toggles between active and inactive states that correlate with opening and closing of the catalytic cleft (Figs 8 and S14). MLi-2, a type I kinase inhibitor, locks the kinase domain into a closed and active-like conformation, while a type II inhibitor is hypothesized to lock the kinase domain into an open conformation [10]. The structure of monomeric LRRK2$_{RCKW}$ serves as a model for the inactive kinase, while the full-length I2020T LRRK2 mutant docked onto microtubules in a helical manner represents an active dimer [9]. Opening and closing of the kinase cleft, where the N- and C-lobes move as rigid bodies, is determined by the flexibility of the N-lobe and its ability to interact with the C-lobe. In its inactive state, it is locked into an open conformation by the 2 flanking helices, the COR-B Helix in COR-B domain and the Ct-Helix as well as by an unusual DFGψ motif in LRRK2, where the highly conserved Phe is replaced with Tyr (DYGψ) [8]. PD mutations that lead to activation obviously alter the equilibrium between the active and inactive states. We showed previously how the 2 PD mutations in the kinase domain (G2019S and I2020T) release the inhibitory NTDs, and with our HDX-MS analysis of the kinase domain, we showed how the disordered region surrounding the AS becomes more ordered by the binding of MLi-2 [12].

In addition to mediating allosteric communication between the kinase domain and the GTPase (ROC) domain, the COR-B domain also controls dimerization of the active kinase [9,11]. Embedded within COR-B are 2 domain:domain interfaces (Fig 8). The COR-B Helix interface binds directly with the αC helix in the kinase domain, while the COR-B:ROC interface is sensitive to the conformation of the ROC domain as well as PD-related mutations. We hypothesize that these mutations (R1441 and N1437 in the ROC domain and Y1699 in COR-B domain) also release the inhibitory NTDs by destabilizing this COR-B:ROC interface [8,30]. One final mutation at this interface (R1398H/K) is actually a protective mutation in both PD and inflammatory bowel disease (IBD) [31]. A final feature of the LRRK2$_{RCKW}$ cryo-EM structure is a bound GDP and T1343 is phosphorylated [30]. Whether this is physiologically important remains to be established; however, T1343 is homologs G12 in RAS and is common disease mutation [32,33].

Communication between the kinase and GTPase domains in LRRK2 is mediated primarily by the domain:domain interfaces of the COR-B domain, while direct contact between the kinase domain and COR-A is controlled by the hinging motion of the kinase domain (Fig 8). Dimerization is also mediated by the COR-B domain as predicted by Watanabe and colleagues and validated by the structure of full-length LRRK2 [9,11]. The 2 clusters of PD mutations highlight the importance of the COR-B:ROC domain interface and the hinging motion of the kinase domain, while all of the mutations potentially break apart the inhibition that is imposed by the NTDs. The dominant organizing motif in COR-B is the COR-B Helix, while the dominant motif in the kinase domain is the DYGψ motif as described previously [12]. With HDX-MS and GaMD, we are beginning to achieve a deeper molecular understanding of these critical domain:domain interfaces as well as loop dynamics, which all contribute to the allosteric regulation of LRRK2 and are perturbed by mutations that make LRRK2 a risk factor for PD.

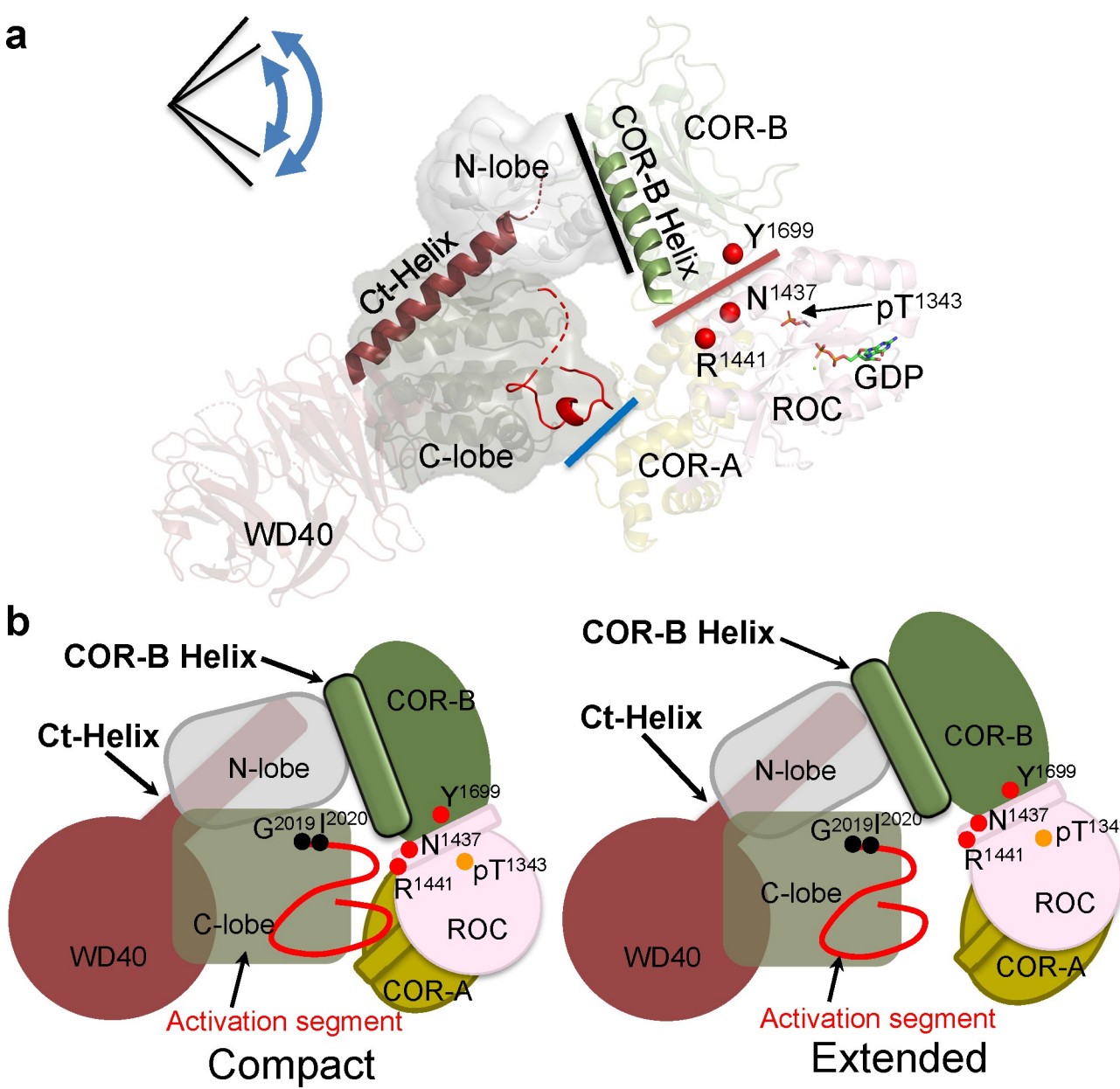

**Fig 8. The interfaces in compact or extended conformation of LRRK2$_{RCKW}$.** (a) The COR-B Helix and Ct-Helix are highlighted on the cryo-EM structure of LRRK2$_{RCKW}$. The pathogenic mutations N1437, R1441, and Y1699 are shown as red spheres. The lines show the domain:domain interfaces: COR-A domain:C-lobe (blue); ROC domain:COR-B domain (red); COR-B domain:N-lobe (black). The kinase domain toggles between open and closed conformations that lead to the compact or extended states of LRRK2$_{RCKW}$. (b) Cartoon representation of the compact and extended states of LRRK2$_{RCKW}$. The interaction between the COR-B domain and the N-lobe of the kinase domain, and the ROC domain as well as the COR-B domain remain intact when the COR-A domain moves away from the C-lobe of the kinase in the extended conformation. cryo-EM, cryogenic electron microscopy; Ct-Helix, C-terminal helix; ROC, ras-of-complex.

## Materials and methods

### Hydrogen–deuterium exchange mass spectrometry

LRRK2$_{RCKW}$ proteins were expressed and purified from Sf9 cell [12]. HDX-MS was performed using a Waters Synapt G2Si equipped with nanoACQUITY UPLC system with H/DX technology and a LEAP autosampler. The LRRK2$_{RCKW}$ concentration was 5 μM in LRRK2 buffer

containing 20 mM HEPES/NaOH (pH 7.4), 800 mM NaCl, 0.5 mM TCEP, 5% Glycerol, 2.5 mM $MgCl_2$, and 20 μM GDP. The deuterium uptake was measured in LRRK2 buffer in the presence and absence of the kinase inhibitor MLi-2 (50 μM). For each deuteration time, 4 μL complex was equilibrated to 25°C for 5 min and then mixed with 56 μL $D_2O$ LRRK2 buffer for 0, 0.5, 1, or 2 min. The exchange was quenched with an equal volume of quench solution (3 M guanidine, 0.1% formic acid (pH 2.66)). The quenched sample (50 μL) was injected into the sample loop, followed by digestion on an in-line pepsin column (immobilized pepsin; Pierce) at 15°C. The resulting peptides were captured on a BEH C18 Vanguard precolumn, separated by analytical chromatography (Acquity UPLC BEH C18, 1.7 μM, 1.0 × 50 mm, Waters Corporation) using a 7% to 85% acetonitrile gradient in 0.1% formic acid over 7.5 min, and electrosprayed into the Waters SYNAPT G2Si quadrupole time-of-flight mass spectrometer. The mass spectrometer was set to collect data in the Mobility, ESI+ mode; mass acquisition range of 200 to 2,000 (m/z); scan time 0.4 s. Continuous lock mass correction was accomplished with infusion of leu-enkephalin (m/z = 556.277) every 30 s (mass accuracy of 1 ppm for calibration standard). For peptide identification, the mass spectrometer was set to collect data in $MS^E$, ESI+ mode instead.

The peptides were identified from triplicate $MS^E$ analyses of 10 μM LRRK2$_{RCKW}$, and data were analyzed using PLGS 3.0 (Waters Corporation). Peptide masses were identified using a minimum number of 250 ion counts for low energy peptides and 50 ion counts for their fragment ions. The peptides identified in PLGS were then analyzed in DynamX 3.0 (Waters Corporation) using a cutoff score of 6.5, error tolerance of 5 ppm, and requiring that the peptide be present in at least 2 of the 3 identification runs. The peptides reported on the coverage maps are those from which data were obtained. The relative deuterium uptake for each peptide was calculated by comparing the centroids of the mass envelopes of the deuterated samples versus the undeuterated controls [34]. For all HDX-MS data, at least 2 biological replicates were analyzed each with 3 technical replicates. Data are represented as mean values +/− SEM of 3 technical replicates due to processing software limitations; however, the LEAP robot provides highly reproducible data for biological replicates. The deuterium uptake was corrected for back exchange using a global back exchange correction factor (typically 25%) determined from the average percent exchange measured in disordered termini of various proteins [35]. Deuterium uptake plots were generated in DECA (github.com/komiveslab/DECA), and the data are fitted with an exponential curve for ease of viewing [36].

## Gaussian accelerated molecular dynamics (GaMD) simulation

The LRRK2$_{RCKW}$ model for simulations were prepared based on the reported LRRK2$_{RCKW}$ structure (PDB: 6VP6) using Modeller to model the missing loops [37]. The Protein Preparation Wizard was used to build missing sidechains and model charge states of ionizable residues at neutral pH. Hydrogens and counter ions were added, and the models were solvated in a cubic box of TIP4P-EW water [38] and 150 mM KCl with a 10-Å buffer in AMBER tools D.A. Case, 2016 #731}. AMBER16 was used for energy minimization, heating, and equilibration steps, using the CPU code for minimization and heating and GPU code for equilibration. Parameters from the Bryce AMBER parameter database were used for phosphoserine and phosphothreonine [39]. Systems were minimized by 1,000 steps of hydrogen-only minimization, 2,000 steps of solvent minimization, 2,000 steps of ligand minimization, 2,000 steps of side-chain minimization, and 5,000 steps of all-atom minimization. Systems were heated from 0 K to 300 K linearly over 200 ps with 2 fs time-steps and 10.0 kcal/mol/Å position restraints on protein. Temperature was maintained by the Langevin thermostat. Constant pressure equilibration with an 8-Å nonbonded cutoff with particle mesh Ewald was performed with 300 ps of protein and peptide restraints followed by 900 ps of unrestrained equilibration. GaMD was

used on GPU-enabled AMBER16 to enhance conformational sampling [40]. GaMD applies a Gaussian-distributed boost energy to the potential energy surface to accelerate transitions between metastable states while allowing accurate reweighting with cumulant expansion. Both dihedral and total potential acceleration were used simultaneously. Potential statistics were collected for 2 ns followed by 2 ns of GaMD during which boost parameters were updated for each simulation. Each GaMD simulation was equilibrated for 10 ns. For each construct, 10 independent replicates of 200 ns of GaMD simulation were run in the NVT ensemble, for an aggregate of 2.0 μs of accelerated MD.

## Supporting information

**S1 Fig. The deuterium uptake of LRRK2 in apo and MLi-2–bound conditions.** LRRK2, leucine-rich repeat kinase 2; ROC, ras-of-complex.
(PDF)

**S2 Fig. The deuterium uptake of LRRK2$_{RCKW}$.**
(PDF)

**S3 Fig. LRRK2$_{RCKW}$ is fluctuated between compact and extended conformation.** PMF, potential of mean force.
(PDF)

**S4 Fig. The deuterium uptake of the kinase domain is influenced by the domains that flank the kinase domain of LRRK2.** A-loop, activation loop; Ct-Helix, C-terminal helix; LRRK2, leucine-rich repeat kinase 2; ROC, ras-of-complex.
(PDF)

**S5 Fig. Interactions between A-loop and COR-B Helix as shown with GaMD simulations.** A-loop, activation loop; GaMD, Gaussian accelerated molecular dynamics.
(PDF)

**S6 Fig. Deuterium uptake around the Ct-Helix.** Ct-Helix, C-terminal helix.
(PDF)

**S7 Fig. The interactions of pT2524.**
(PDF)

**S8 Fig. AS of PKA.** AS, activation segment.
(PDF)

**S9 Fig. Interfaces created by the A-loop and the APE-αF linker between the catalytic and regulatory subunits of PKA (PDB: 2QCS).** A-loop, activation loop.
(PDF)

**S10 Fig. AS of inactive full-length LRRK2 aligned with active PKA.** AS, activation segment; GaMD, Gaussian accelerated molecular dynamics; LRRK2, leucine-rich repeat kinase 2.
(PDF)

**S11 Fig. The APE motif is anchored to the αH-αI loop.** cryo-EM, cryogenic electron microscopy; LRRK2, leucine-rich repeat kinase 2; MD, molecular dynamics.
(PDF)

**S12 Fig. A-loop is anchored by COR-A and COR-B domains.** A-loop, activation loop; LRRK2, leucine-rich repeat kinase 2; MD, molecular dynamics; ROC, ras-of-complex.
(PDF)

**S13 Fig. Three rigid bodies of LRRK2$_{RCKW}$.** Ct-Helix, C-terminal helix; ROC, ras-of-complex.
(PDF)

**S14 Fig. Domain contacts highlights the Ct-Helix interactions.** A-loop, activation loop; Ct-Helix, C-terminal helix; ROC, ras-of-complex.
(PDF)

**S1 Data. HDX-MS data.** HDX-MS, hydrogen–deuterium exchange mass spectrometry.
(XLSX)

**S1 Movie. Snapshots from MD simulation of LRRK2$_{RCKW}$ showing the AS.** AS, activation segment; MD, molecular dynamics.
(AVI)

**S1 File. Pymol file for Fig 4B.**
(PSE)

**S2 File. Pymol file for Fig 4E.**
(PSE)

**S3 File. Pymol file for Fig 6B.**
(PSE)

**S1 Table. HDX-MS data.** HDX-MS, hydrogen–deuterium exchange mass spectrometry.
(XLSX)

## Author Contributions

**Conceptualization:** Jui-Hung Weng, Phillip C. Aoto, Robin Lorenz, Friedrich W. Herberg, Susan S. Taylor.

**Data curation:** Jui-Hung Weng, Phillip C. Aoto, Steve Silletti, Sebastian Mathea, Susan S. Taylor.

**Formal analysis:** Jui-Hung Weng, Phillip C. Aoto, Robin Lorenz, Jian Wu, Steve Silletti, Susan S. Taylor.

**Funding acquisition:** Friedrich W. Herberg, Susan S. Taylor.

**Investigation:** Jui-Hung Weng, Phillip C. Aoto.

**Methodology:** Jui-Hung Weng, Phillip C. Aoto, Steve Silletti.

**Project administration:** Friedrich W. Herberg, Susan S. Taylor.

**Resources:** Sebastian Mathea, Deep Chatterjee.

**Supervision:** Susan S. Taylor.

**Validation:** Jui-Hung Weng, Phillip C. Aoto.

**Visualization:** Jui-Hung Weng, Phillip C. Aoto, Jian Wu, Susan S. Taylor.

**Writing – original draft:** Jui-Hung Weng, Phillip C. Aoto, Robin Lorenz, Jian Wu, Pallavi Kaila-Sharma, Susan S. Taylor.

**Writing – review & editing:** Jui-Hung Weng, Phillip C. Aoto, Robin Lorenz, Jian Wu, Sven H. Schmidt, Jascha T. Manschwetus, Pallavi Kaila-Sharma, Sebastian Mathea, Stefan Knapp, Friedrich W. Herberg, Susan S. Taylor.

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
