## [Editor Report · Decision Letter 0]

22 Sep 2021

Dear Susan, 

Thank you for submitting your manuscript entitled "Two helices control the dynamic crosstalk between the catalytic domains of LRRK2" for consideration as a Research Article by PLOS Biology.

Your manuscript has now been evaluated by the PLOS Biology editorial staff as well as by an academic editor with relevant expertise and I am writing to let you know that we would like to send your submission out for external peer review.

Please re-submit your manuscript within two working days, i.e. by Sep 24 2021 11:59PM.

Kind regards,

Ines

--

Ines Alvarez-Garcia, PhD

Senior Editor

PLOS Biology

---

## [Decision Letter · Decision Letter 1]

12 Nov 2021

Dear Susan,

Thank you for submitting your manuscript entitled "Two helices control the dynamic crosstalk between the catalytic domains of LRRK2" for consideration as a Research Article at PLOS Biology. Thank you also for your patience as we completed our editorial process, and please accept my apologies for the delay in providing you with our decision. Your manuscript has been evaluated by the PLOS Biology editors, an Academic Editor with relevant expertise, and by three independent reviewers.

You will see that the reviewers find the conclusions of your study interesting and the manuscript worth pursuing for publication, however they also raise several points that would need to be addressed. Reviewer 2 thinks that including HDX data for the PD mutation R1441G/C/H would add significantly to the paper, but after discussing this point with the Academic Editor we have decided not to make it a requirement for publication considering the time that it is likely to require. Nevertheless, if you happen to have this data available, we agree with the reviewer that it would be a great addition to the manuscript. Reviewer 3 also requests a comparison to the AlphaFold2 structure and we do think that this information would add interest to the study.

In light of the reviews (attached below), we are pleased to offer you the opportunity to address the comments from the reviewers in a revised version that we anticipate should not take you very long. We will then assess your revised manuscript and your response to the reviewers' comments and we may consult the reviewers again.

We expect to receive your revised manuscript within 1 month.

**IMPORTANT - SUBMITTING YOUR REVISION**

3. Resubmission Checklist

a) *Published Peer Review*

b) *PLOS Data Policy*

d) *Blurb*

Please also provide a blurb which (if accepted) will be included in our weekly and monthly Electronic Table of Contents, sent out to readers of PLOS Biology, and may be used to promote your article in social media. The blurb should be about 30-40 words long and is subject to editorial changes. It should, without exaggeration, entice people to read your manuscript. It should not be redundant with the title and should not contain acronyms or abbreviations. For examples, view our author guidelines: https://journals.plos.org/plosbiology/s/revising-your-manuscript#loc-blurb

Sincerely,

Ines

--

Ines Alvarez-Garcia, PhD

Senior Editor

PLOS Biology

Reviewers’ comments

Rev. 1:

The manuscript "Two helices control the dynamic crosstalk between the catalytic domains of LRRK2" by Weng et al uses (HDX-MS) and Molecular Dynamics (MD) simulations to explore the dynamics and activation of LRRK2, a key kinase in PD pathogenesis. This study, based on a deep understanding of kinase activation, provided extremely valuable information underlying the working mechanism of LRRK2 and pointed out rigid bodies as well as dynamic interfaces upon activation. The work is well-designed and experiments are logically performed. I only have few minor points:

1) Line 99 and Line 101

For clarification, it will nice if authors can be more specific about sequences of the "two helices of the COR-A domain at the interface contacting the ROC domain" and "the ROC helix and the adjacent loops of ROC-B domain"

2) line 182-183: "an active kinase the basic residue that lies at the beginning of the αC helix (R1915 in LRRK2) interacts with a phosphate on the A-Loop". Please clarify which "phosphate" of A-loop here could interact with R1915 if possible?

3) line 256-258 ""The mainly negatively charged residues on the other surface could potentially create a binding interface with the N-terminal domains (NTDs) of LRRK2 when it is inhibited".

Here the reviewer assumes that "inhibited" means "when MLi2 is bound".

4) line 267-268: COR-B-Kinase linker (residue 1840-1861 and residue 1862-1876) have 70% and 75% uptake of deuterium at 2 min, respectively.

It will be nice if the authors can label these residue ranges on the figure.

5) regarding the scale of electrostatic surface (Fig. 3b and 5b), what is the unit for the calculation? If it is kT/e, then the number is unusually high (FYI, 1kT/e=25.7 mV). Using APBS plugin in Pymol should give a more reasonable number.

6) Fig. 3b and 3c: the figure legend of 3b is for figure 3c, and vice versa.

7) Fig. 7b: the position of red cycle and neighboring black cycle is shifted by one residue.

Rev. 2:

Weng et al. describe the interplay between Roc, COR-A/COR-B and the kinase domain by Hydrogen-deuterium exchange experiments and MD simulations using a LRRK2 C-terminal four domain construct RCKW, whose protein structure has recently been solved at high resolution by Cryo-EM (Deniston et al.). The HDX experiments as well as the MD simulations are methodologically sound and give valuable mechanistic insights into the intramolecular regulation of LRRK2.

Major concerns:

Nevertheless, it would significantly increase the impact of the work if at least some of the conclusions could be biochemically confirmed, e.g. the finding that the terminal residue E2527 interacts with the COR-B residues R1771 and K1772 within the "Dk-helix" (page 10) as well the conclusion that the PD mutation site R1441 interacts with the residues E1780 and W1791 which keeps the Dk helix in place (page 11).

I am aware that a mutational analysis of all interfaces described in this paper would certainly be very laborious and is likely beyond the scope of this work. Nevertheless, it would be highly appreciated if at least HDX data for the PD mutation R1441G/C/H is provided.

In addition, it would be interesting to see, if the key findings are also in agreement with the recently published full-length structure of LRRK2 (reference 11). The latter not only contains the N-terminal domains but also the three terminal residues found to be disordered in the RCKW structure (page 10, 193).

Minor concerns:

p6 L109/110: "This explains why a stable and active kinase domain of LRRK2 has not been expressed yet" I agree that interaction of the WD40 and kinase can be a reason, but I doubt that it is the only one.

Rev. 3:

This article presents detailed experimental data and molecular dynamics simulations on the intramolecular interactions of structural elements of the LRRK2 protein kinase. The intramolecular dynamics of LRRK2 are important for understanding the effects of LRRk2 mutations observed in Parkinson's disease patients. The HDX data are highly informative and the MD simulations provide some additional insights on the data. On the whole, this is a very valuable study.

I have comments mostly on presentation.

Occasionally the paper gets bogged down in details that are hard to parse. One thing to improve this would be consistency in mentioning which residues are being discussed (e.g. line 194 "a flexible loop" could list the residues). When different PDB structures are discussed, the PDB codes should be given (multiple times if necessary if going back and forth between them).

The word (and cliché) "crosstalk" is overused in this paper (as it is in many cell biology papers); exactly what is meant by it is unclear. Does it mean interaction or correlated motions or allosteric structural or dynamic effects? It should be defined and in many places it could be replaced with more precise language, depending on the sense that is meant. The same is true of the word "unleashed." There could be some data from the simulations on the distribution of distances between residues in the C-terminal domain of the kinase with residues in the the Cor-A domain (from compact to cryo-EM to extended, as in Figure 1). In some places, PD mutations are described as "unleashing" domains from each other. A simulation that shows this would add a great deal to the paper.

Other anthropomorphizing words are also used and the meaning is sometimes unclear, e.g "communicates", line 157.

Papers with MD simulations are sometimes difficult to interpret since the reader is restricted to looking at the protein structure images provided by the authors. It would be good for the authors to provide coordinates and Pymol or Chimera sessions for the MD snapshots that they show in figures. I don't think it's useful or necessary to provide the whole trajectory (nobody else wants to analyze all that). But coordinates for the important snapshots in the paper are critical for people to be able to analyze the structural insights presented in the paper. Static images are not enough for this.

The Alphafold2 predicted structure of LRRK2 may be useful to compare to the experimental structures and the simulations.

https://alphafold.ebi.ac.uk/files/AF-Q5S007-F1-model_v1.cif

This structure is an active "BLAminus" structure, while the cryo-EM structures are mostly "SRC-inactive" or BLBplus:

http://dunbrack.fccc.edu/kincore/GENE/LRRK2

The Alphafold2 structure provides the full activation segment in an active form. It also provides coordinates for loops missing in the cryo-EM structure. Some predicted interactions that are missing in the cryo-EM but were present in the simulations (and experimental structures of other kinases) are present in the Alphafold structure (e.g. line 326, the interaction between E2042 and R2122). It may provide very valuable information for the analysis presented in this paper, which should be considered (it is a prediction of course, but no more so than the MD simulations that the authors use to make inferences about structure-function relationships in LRRK2).

On line 181, the authors say the cryo-EM structure (presumably 6VP6 chain A) is in a C-helix-out conformation. It does not look like that way to me. Compare it to active BLAminus structure of BRAF, PDB 6UAN chain C by aligning the N-terminal domains. Compare the BLBplus (SRC-inactive) structure of BRAF in PDB 7M0V, chain A.

Minor issue.

Reference 10 is incomplete. It's a key reference in the paper. Other references missing page numbers: 8,11,12

---

## [Editor Report · Decision Letter 2]

17 Dec 2021

Dear Susan,

Thank you for submitting your revised Research Article entitled "Two helices control the dynamic crosstalk between the catalytic domains of LRRK2" for publication in PLOS Biology. I have now obtained advice from the original Academic Editor and have discussed these comments with the editorial team. 

Based on this, we will probably accept this manuscript for publication, provided you satisfactorily address the remaining data and other policy-related requests. In addition, we would like you to consider a suggestion to improve the title:

"LRRK2 dynamics analysis identifies allosteric control of the crosstalk between its catalytic domains"

We expect to receive your revised manuscript within two weeks. 

*Published Peer Review History*

*Early Version*

Sincerely,

Ines

--

Ines Alvarez-Garcia, PhD,

Senior Editor,

ialvarez-garcia@plos.org,

PLOS Biology

Fig. 2A; Fig. 5 and Fig. S6A

---

## [Editor Report · Decision Letter 3]

14 Jan 2022

Dear Susan,

On behalf of my colleagues and the Academic Editor, Kylie Walters, I am pleased to say that we can in principle accept your Research Article entitled "LRRK2 dynamics analysis identifies allosteric control of the crosstalk between its catalytic domains" for publication in PLOS Biology, provided you address any remaining formatting and reporting issues. These will be detailed in an email that will follow this letter and that you will usually receive within 2-3 business days, during which time no action is required from you. Please note that we will not be able to formally accept your manuscript and schedule it for publication until you have any requested changes.

PRESS

Sincerely, 

Ines

--

Ines Alvarez-Garcia, PhD 

Senior Editor 

PLOS Biology
